# A Review of the Impacts and Opportunities for African Urban Dragonflies

**DOI:** 10.3390/insects12030190

**Published:** 2021-02-24

**Authors:** Charl Deacon, Michael J. Samways

**Affiliations:** Department of Conservation Ecology and Entomology, Stellenbosch University, Matieland, Stellenbosch 7600, South Africa; samways@sun.ac.za

**Keywords:** aquatic insects, biodiversity conservation, Cape Floristic Region, community engagement, Highveld, Maputaland-Pondoland-Albany biodiversity hotspot, mitigation, Odonata, southern Africa

## Abstract

**Simple Summary:**

The expansion of urban areas in combination with climate change places great pressure on species found in freshwater habitats. Dragonflies are iconic freshwater organisms due to their large body sizes and striking coloration. They have been widely used to indicate the impacts of natural and human-mediated activities on freshwater communities, while also indicating the mitigation measures required to ensure their conservation. Here, we review the major threats to dragonflies in southern Africa, specifically those in urban areas. We also provide information on effective mitigation measures to protect dragonflies and other aquatic insects in urban spaces. Using three densely populated areas as case studies, we highlight some of the greatest challenges for dragonflies in South Africa. More importantly, we give a summary of current mitigation measures which have maintained dragonflies in urban spaces. In addition to these mitigation measures, public involvement and raising awareness contribute greatly to the common cause of protecting dragonflies around us.

**Abstract:**

Urban settlements range from small villages in rural areas to large metropoles with densely packed infrastructures. Urbanization presents many challenges to the maintenance of freshwater quality and conservation of freshwater biota, especially in Africa. There are many opportunities as well, particularly by fostering contributions from citizen scientists. We review the relationships between dragonflies and urbanization in southern Africa. Shifts in dragonfly assemblages indicate environmental change, as different species are variously sensitive to abiotic and biotic water and bank conditions. They are also conservation umbrellas for many other co-occurring species. Major threats to southern African dragonflies include increasing infrastructure densification, frequent droughts, habitat loss, pollution, and invasive alien vegetation. Mitigation measures include implementation of conservation corridors, maintenance of healthy permanent ponds, pollution reduction, and removal of invasive alien trees. Citizen science is now an important approach for supplementing and supporting professional scientific research.

## 1. Introduction

Human settlements have increased in global proportion, especially during the last century [1]. Urban settlements range from villages with fuzzy boundaries in rural areas to large cities with dense human populations. Over time, villages growing into towns and towns growing into cities result in rapid infrastructural densification [2,3]. While large cities are becoming more prevalent, classification boundaries between types of settlements are often unclear [4]. Yet, urban settlements are generally hardscapes overlying greenscapes [5]. Gradients of urbanization have complex consequences for the global natural environment [6,7]. These impacts broadly include habitat loss, pollution, alien vegetation encroachment, and changes in microclimates, and their effects often extend far beyond settlement centers [8,9,10].

Urban landscapes are not usually designed for biodiversity conservation [11]. A manifestation of this is that insect species richness generally declines towards city centers [12,13]. However, urban greening can be highly beneficial for insect biodiversity [14]. This involves including green spaces, such as botanical gardens, public parks and cemeteries in urban design, along with clearing alien vegetation and encouraging native vegetation [5]. These green spaces allow insects from a range of functional groups to persist in areas dominated by urbanization, while simultaneously promoting human well-being and bringing urbanites closer to nature.

Africa is the fastest urbanizing continent [15], with settlements being established close to waterbodies in response to severe water scarcity, especially across sub-Saharan Africa where El Niño events lead to unstable water supplies [16]. Due to the space limitations in existing urban areas, the fastest urban growth is in small towns with high poverty levels. Ecological impacts are expected to be particularly severe in ecologically sensitive areas [17,18], further exacerbated by ineffective environmental regulations due to limited scientific knowledge on biological responses to urbanization [15]. Africa presents a range of abiotic and biotic challenges to biodiversity, and with much of the continent still undeveloped, urbanization is considered one of the greatest looming threats.

Within sub-Saharan Africa, southern Africa is the most urbanized region due to its accessible coastline, abundant mineral deposits and rich soils [19]. In southern African cities, urbanization primarily increases as a result of rural-urban migration. In contrast, local human population growth is the leading driver of urbanization in rural areas, albeit slower compared to major cities. Yet, scientists have been slow to investigate the ecological impact of urbanization on the rich African insect fauna. Most aquatic insect groups are poorly studied across the continent [20]. Southern African dragonflies (referring to both dragonflies and damselflies; Odonata: Anisoptera and Zygoptera) are by far the most studied group of insects in terms of their taxonomy and biogeography [21].

Here, we review the interface between dragonflies and urban/suburban areas across southern Africa. We provide insight into the direct and indirect impacts of urbanization on dragonflies in the global context, and highlight how certain urban areas provide particular opportunities for dragonfly occupancy and conservation. Using three densely-populated regions in water-scarce South Africa as case studies, we also synthesize urban impacts and opportunities for dragonflies. Finally, we evaluate citizen science as a data collection method for investigating and monitoring urban dragonfly ecology, and make recommendations for future research.

## 2. Urban Impacts on Dragonflies and Their Habitats

### 2.1. Direct Impacts on Dragonflies

Urbanization places immense direct pressure on freshwater habitats (Figure 1). Infilling and draining for infrastructure development are a major cause of freshwater habitat destruction [22,23], with losses in freshwater habitats strongly correlated with losses of green space in some countries [24]. About 50% of global ponds and wetlands have been lost during the last century, partly due to urbanization [25]. Furthermore, global urbanization during the last 50 years has led to sediment loads in rivers being up to 120 times higher than natural loads [26]. African pond and river losses have been poorly documented, yet it is estimated that as much as 97% of all wetlands have been lost around Cape Town, South Africa, as a direct result of infrastructural development [27]. These numbers reflect the potential wide impacts of accelerating urbanization across Africa.

Translating habitat losses to biodiversity losses is challenging, since information on the historical states of freshwater habitats across Africa is lacking. Nearly 6% of all described insect species occupy freshwater habitats [28], including over 900 species of dragonflies in sub-Saharan Africa [29]. These dragonflies and other aquatic insects are under immense pressure from accelerating rates of urbanization at the expense of natural waterbodies, ultimately leading to decreasing population sizes and local extinctions [30]. Habitat loss is particularly problematic for rare species that occupy natural ponds. Although these ponds are often regarded an ecologically unimportant, they are in fact highly complementary at the regional scale [31].

High road density is almost synonymous with urbanization, and roads affect dragonflies in several ways (Figure 1) [24]. Roads can limit movement between favorable habitats and act as dragonfly movement barriers [32], resulting in progressive isolation of populations and loss of genetic diversity over time. Roads and other glossy surfaces may also attract some flying insects, especially adult dragonflies, functioning as ecological traps so placing additional pressure on breeding populations, especially for localized species [33]. Although no information is available for Africa, >60% of all insects killed by moving vehicles in India are dragonflies [34], while in Japan the figure is only 17% [35]. These differing figures suggest that various factors are involved, ranging from vehicle density and speed, to the type of roadside environment, to the distance from a waterbody [24]. Even dragonfly flight behavior has an effect, with species that fly close to the ground being impacted the most [36]. These findings indicate that road construction close to dragonfly habitats should be avoided, especially important sites for endemic species [37]. 

### 2.2. Indirect Impacts on Dragonflies

Indirect urban impacts on aquatic insects are complex, and their collective magnitude may outweigh the magnitude of direct impacts [24]. Many adult aquatic insects, including dragonflies, rely on terrestrial environments. Urban modification of their terrestrial surroundings is as important as modifications to their freshwater habitats, and has far-reaching effects on their fitness (Figure 1). For example, many dragonfly species require submerged or overhanging marginal vegetation for oviposition. Emergent adults (tenerals) also move away from water to vegetation, only returning to aquatic habitats when mature [38]. The loss of terrestrial vegetation, thus, reduces functional connectivity, limits local dragonfly movement and inhibits natural dragonfly population dynamics. Indeed, vegetation cover is often a better predictor of dragonfly occupancy than in-water conditions, and vegetation conservation and restoration can even buffer the effects of urbanization [39,40]. Dragonflies in general respond more to vegetation cover and complexity than to plant species identity. Yet, some do not rely on vegetation to complete their life cycles, meaning that preservation of vegetation integrity alone does not guarantee dragonfly occupancy.

On the other hand, when marginal vegetation is too dense, there is a decline in dragonfly diversity (Figure 1) [41,42]. When dense stands of invasive alien shrubs and trees, especially *Acacia* trees, replace native riparian vegetation in urban areas, dragonfly habitats are adversely transformed mainly through loss of sunny local environments [43,44]. About a quarter of all South African dragonflies are endemic to the country, with twelve red-listed as threatened [45]. These dragonflies are almost exclusively threatened by alien vegetation in their habitats. In addition to shading, alien vegetation contributes to soil erosion through homogenization of riparian plant communities, and leads to increased runoff into rivers and higher sediment loads [46]. Invasive water hyacinth (*Eichhornia crassipes*) is another cause of concern across Africa, forming dense mats of floating vegetation on ponds and lakes in urban areas and some protected sites [47]. Riparian and floating invasive plants eliminate native species and reduce overall habitat heterogeneity (e.g., through a reduction in oxygen supplies and increases in water temperature), which in turn leads to local changes from specialist dragonfly assemblages to assemblages dominated by generalists [48].

In arid and semi-arid countries, construction of dams alleviates water scarcity for the benefit of agricultural, industrial and domestic sectors [21]. Damming does not destroy aquatic habitats so much as converts lotic (flowing) into lentic (standing) habitats, with a concurrent shift in dragonfly composition from a characteristically lotic assemblage to a lentic one, as shown for the subtropical region of South Africa [49]. Combined with alien vegetation, damming of rivers can lead to great reductions in macroinvertebrate diversity (Figure 1) [50]. River impoundment for water security leads to a local loss of almost half of the native riverine dragonfly species, although this recovers farther downstream [51]. Damming of rivers also introduces other disturbances, such as alien fish (e.g., *Oncorhynchus mykiss* and *Micropterus salmoides*) for recreational fishing. These generalist predators feed in different areas of the water column, leading to population declines of dragonfly larvae and adults. Other damming disturbances include increased wave action from boating, which can have a great impact on marginal vegetation integrity, and uncontrolled civilian access, leading to bank disturbance and increased pollution levels [5].

Freshwater contaminants predominantly originate from construction, maintenance, production activities, and erosion [52,53,54]. Trace metals and alteration of water temperature, pH, conductivity, and dissolved oxygen gradients are of particular concern [52,55]. Dragonflies are variably sensitive to these characteristics and their occupancy is determined by physicochemical gradients (Figure 1) [56]. While highly polluted urban waterbodies function as ecological traps for dragonflies, some adult dragonflies are able to assess physicochemical properties of aquatic habitats and may avoid breeding in them [57]. Sensitive species may be killed by pollutants and, with less competition, tolerant species may dominate [58,59]. Some widespread species, such as *Ischnura senegalensis*, are adapted to breed in ponds and slow-moving water channels engineered by hippos, and where defecation takes place. These damselflies are therefore predisposed to occupy organically polluted urban water bodies.

Other less obvious impacts of urbanization, such as excessive artificial lighting, may also have an indirect impact on dragonflies (Figure 1). Some dragonfly species may be attracted to artificial lighting sources [60], especially in the case of certain crepuscular species such as those in the genus *Gynacantha*. Artificial lighting may interfere with dragonfly dispersal, and also function as ecological traps through increasing encounter rates with insectivorous predators [61].

Potential urban impacts on dragonflies and other aquatic macroinvertebrates vary with context. For instance, a comparison of urban and rural ponds in the United Kingdom showed that macroinvertebrate assemblage composition varied among pond types, and urban ponds made an equal contribution to regional diversity as rural ponds [62]. Conversely, a long-term comparison between urban stormwater ponds and natural ponds in Canada showed that anisopteran species richness and abundance, and zygopteran abundance, were lower for stormwater ponds, suggesting that urban ponds have little added value in this particular regional context [63]. In South Africa, a comparison between agricultural and urban rivers showed that agricultural and urban land use changes have similar effects on total dragonfly species richness, endemic dragonfly species richness, and zygopteran species richness alone, but not anisopteran species richness [64]. These results suggest that agricultural and urban land use change equally contribute to overall biotic homogenization, but the two dragonfly suborders may respond differently to various types of disturbances.

## 3. Opportunities for Dragonflies in Urban Areas

As much as 80% of land may be covered by hard surfaces in city centers [65], and a rising skyline strongly limits the movement of individuals. Urban green spaces, such as parks, gardens, cemeteries and nature reserves, are fragmented habitat patches in an otherwise inhospitable matrix. These urban green spaces are no substitute for wild areas, but they play an important role as refuges and supplementary habitats, both for widespread species and those of high conservation concern (Figure 2) [5]. Although urban green spaces are potential sites for alien vegetation colonization, this can be circumvented by establishing native vegetation that improves functional connectivity for insect populations [24,39]. Biotope requirements are highly variable among dragonfly species [66] and size, quality and connectivity of urban green spaces are essential for maintaining dragonfly assemblages. Most zygopterans require only 50–300 m of favorable environment around their central habitats [67], while certain anisoperans, such as *Sympetrum depressiusculum*, require foraging areas of about 0.5–1 km around their natal habitats [68]. In turn, highly mobile and migratory species such as *Pantala flavescens* may fly over oceans in search of breeding habitats.

Both natural and artificial freshwater habitats are often abundant in urban green spaces (Figure 2) [69]. Although wetlands today are valued for their aesthetics and richness of biodiversity, in past centuries they were considered as miasmatic breeding sites for disease vectors. Health issues aside, urban wetland habitats, whether natural or artificial, can be of high value for aquatic insects, while simultaneously increasing conservation awareness.

In turn, urban rivers and streams with wide (>30 m) riparian zones, but free of woody alien vegetation, provide sunny microhabitat and promote habitat heterogeneity (Figure 2). These green corridors buffer urban impacts and are attractive breeding sites for some dragonflies. Dragonfly assemblages recover remarkably well when alien vegetation is cleared, and some endemic species recover their area of occupancy and part of their former distribution ranges [70]. Artificial wetlands in urban areas initially attract only common and widespread dragonfly species, but they may also attract rare and localized species once native vegetation has had enough time to establish [71].

Urban ponds may be built for aesthetic enhancement, water storage, managing urban hydrology, and/or for biodiversity conservation. Although insect conservation is rarely the primary aim of pond construction, some artificial water bodies have high secondary value as tiny reserves for aquatic insects in urban environments (Figure 2) [72,73]. Overall, small urban ponds may host higher biodiversity compared to large lakes [24,74], especially when heterogenous ponds are considered collectively [62]. For some central European cities, ponds can increase the area of occupancy for nearly 93% of all dragonflies present in the region, especially Mediterranean species. Acknowledging that some with specialized habitat requirements are excluded from urban environments, some dragonfly species colonizing urban ponds may be of high conservation value [75]. In arid regions of Namibia, dams encourage wetland dragonflies to establish breeding populations in formerly inhospitable environments [76]. Similarly, ponds established for municipal use or conservation of game animals (e.g., water holes) in arid regions of South Africa may attract some generalist dragonflies and extend their areas of occupancy into formerly unoccupied regions [21,77]. It is urban ponds and rivers with gradual margins, with moderate levels of native submerged and riparian vegetation, no pollutant input, and no alien fish, which attract the richest set of dragonfly species [73,78].

## 4. Dragonfly Conservation in Urban Areas of South Africa

Extensive existing urban landcover, and the inevitability of increased urbanization in the future, calls for adaptive responses instead of mitigation measures, *per se* [79]. Ecosystem-based adaptation is a sustainable and cost-effective approach for improving adaptive capacity, building on the premise that the natural environment is the most important input into local economies and human well-being (Figure 3). Ecosystem-based adaptation is especially important for cities in the southern hemisphere, owing to their risk-prone states and limited resources to deal with natural disasters [80].

The first implementation of ecosystem-based adaptation in South Africa was in the coastal city of Durban, where the Durban Metropolitan Open Space System (D’MOSS) action plan was designed. 80,000 ha was set aside to protect the Maputaland-Pondoland-Albany global biodiversity hotspot and to maintain ecosystem service delivery [79,81]. For example, the action plan includes the protection of natural inland wetlands that limit the needs for additional stormwater ponds in urban areas, conservation of intact mangroves that reduce coastal flooding and erosion, and alien vegetation removal that restores hydrological function. Similarly, near-natural riverbanks are included to prevent flash floods, while agro-ecological approaches encourage natural pollinators and pest control. Therefore, protection of these habitats leads to ecosystem service delivery without requiring additional expensive infrastructural development.

In addition to ecosystem-based adaptation in the Durban Metropolitan Area (DMA) having high social and economic value, the conservation and management of natural patches and corridors also have high biodiversity value. Since dragonflies rely on terrestrial and aquatic environments to complete their life cycles, protecting a range of aquatic habitats, as well as natural riparian zones, greatly contributes to dragonfly conservation. Conversion of agricultural areas, such as abandoned sugarcane fields, has also contributed to dragonfly conservation, with former irrigation reserves contributing greatly to local dragonfly diversity (Figure 3a). The DMA is rich in dragonfly species, with more than 70 species (and three national endemics) known from the region [77], including species such as *Phaon iridipennis* which penetrates deep into the urban area. This suggests that land preservation through ecosystem-based adaptation has exceptional value for dragonflies.

Elsewhere in the Maputaland-Pondoland-Albany biodiversity hotspot, freshwater bodies in botanical gardens play a major role in local protection of dragonflies (Figure 3b). A dragonfly conservation pond was designed and constructed along an urban stream in the Pietermaritzburg Botanical Gardens (PMG). The purpose of this artificial pond was to promote aquatic plant conservation, while simultaneously forming an integral part of a dragonfly trail to raise awareness on the ecological importance of dragonflies and their conservation [82]. Dragonfly species richness and abundance increased substantially after the addition of lentic components to the existing lotic components [83]. However, some management was required to maintain optimal conditions for indigenous aquatic plants and dragonflies. Interventions included removal of invasive alien vegetation, clearing dense vegetation to halt vegetation succession, and occasional dredging to manage siltation. The public also responded well to the dragonfly trail, which stimulated production of a dedicated and comprehensive identification guide for the PMG [84].

Biodiversity patterns are complex in Cape Town, a coastal city built around Table Mountain in the heart of the Cape Floristic Region (CFR) [85]. This city deviates from the standard model of urban biodiversity, where diversity decreases towards city centers [12,13]. Instead, Cape Town in one of few cities where biodiversity levels increase towards the central area [86]. Within the boundaries of Cape Town alone, there are about 3400 plant species, of which nearly 200 are endemic to the city [87]. Although total dragonfly species richness is not as high as the subtropical regions of southern Africa (close to 50 species), this Mediterranean area is characterized by about 17 national endemic dragonfly species [45,77]. Among these are regional endemics *Chlorolestes conspicuus*, *Allocnemis leucosticta*, *Elattoneura frenulata*, *Pseudagrion draconis*, *Pseudagrion furcigerum*, *Syncordulia venator*, and *Orthetrum julia capicola*. The disproportionally high number of endemic dragonfly species is the result of this region having had a moderate climate for many millions of years, strong orographic patterns, and many streams, rivers, wetlands and ponds, leading to high levels of environmental heterogeneity.

With the city center falling within the Table Mountain National Park, it has received high levels of conservation action (Figure 3c), partly through the extreme topography limiting urban development [87]. Conserving the natural habitats of Table Mountain is important, as it hosts some of the rarest and most ancient endemic dragonfly species, dating back to 59 my [88]. The only dragonfly which has been extirpated from Table Mountain as a result of urbanization is *Orthetrum rubens*. However, this endemic species occupies other mountainous areas nearby, so has not gone extinct [45]. Yet, lowland freshwater habitats around the mountain are particularly vulnerable, and several streams and wetlands have been lost since the onset of urbanization [89]. The freshwater habitats that remain are under severe pressure through physical modification and high nutrient and pollutant inputs [90]. Deteriorated environmental conditions is problematic for dragonflies, and only a handful of species are capable of occupying highly impacted freshwater habitats. Some of the largest remaining wetlands within protected lowland areas have deteriorated greatly, and these low quality urban freshwater habitats have little value for the endemic dragonflies in the CFR (Figure 3d) [58]. Water contamination in the region is particularly difficult to manage, and effective freshwater habitat restoration requires catchment-wide approaches to supplement local mitigation measures.

Inland cities in heavily urbanized regions such as the Tshwane Metropolitan Area in the Gauteng Province of South Africa, an area moderately rich in dragonfly species (just over 40 species), have many artificial ponds and lakes for water storage used in mining and agricultural activities (Figure 3e) [91]. Tailing ponds were specifically established to store byproducts of mining activities, and when mining was stopped during the 1970s, little environmental legislation was in place to enforce rehabilitation of tailing ponds [92]. Due to the overall high levels of pollution associated with mining, ponds and rivers are mostly occupied by widespread and generalist dragonfly species [93,94]. Although rehabilitation of impacted ponds and rivers is challenging, liming and leaching have been effective in encouraging marginal vegetation growth [92], an important step towards encouraging dragonflies and other aquatic insects. Artificial wetland construction has also been effective in passively assimilating heavy metals in freshwater ecosystems, while simultaneously raising aesthetic appeal of heavily impacted watercourses [95]. During the development of Tshwane’s suburbs in the mid-1950s, some streams and rivers which flow through urban areas were recognized in efforts to preserve green and blue spaces, while acknowledging that ‘wild’ areas have higher value to citizens compared to intensively managed parks and greenways (Figure 3f) [96]. However, rehabilitated and/or preserved green and blue spaces have the potential to improve functional connectivity among existing nature reserves and other open spaces, and have the potential to attract high levels of biodiversity [97]. Indeed, several dragonflies, including some South African endemic species such as *Africallagma sapphirinum* and *Proischnura rotundipennis*, benefit greatly from artificial ponds in heavily urbanized environments [45].

## 5. Dragonflies as Sentinels for Evaluating Urban Impacts through Citizen Science

The relationships between dragonfly occupancy, ecological integrity and urbanization are still poorly understood in Africa. One of the greatest challenges is that historical data for dragonfly occupancy before the onset of urbanization do not exist or are very limited. This further restricts thorough understanding of how dragonfly assemblages and population sizes have changed over time in response to urbanization. However, dragonflies are valuable as sentinels for an array of environmental disturbances, including climate change and habitat degradation [45,98]. They are valuable indicators of urban impact, and may help inform decisions regarding future urban development. They can also help identify management and restoration needs of freshwater habitats in urban areas. Furthermore, dragonflies are effective conservation umbrellas, meaning that dragonfly conservation also benefits other co-occurring freshwater taxa [99,100].

Dragonflies have large body sizes and striking male coloration (Figure 4). They are highly charismatic and iconic among freshwater organisms, making them one of few non-pollinator insect groups with high public appeal [101]. These characteristics make dragonflies ideal candidates for citizen science initiatives, simultaneously promoting the socioecological and psychological aspects of insect conservation [102]. Dragonfly citizen science also creates unique opportunities to rapidly generate large dragonfly distribution and abundance datasets through collaborations between scientists and the broader public. Although citizen science project may be difficult in rural areas with small, dispersed human populations, they can be effective in urban areas. The comparatively rapid nature of citizen science data collection against professional scientific data collection is also useful from a temporal perspective, where regular data collection across wide urban areas enables ecologists to model dragonfly responses to ongoing urbanization and/or restoration.

Many lessons can be learned from the successful citizen science projects in southern Africa, where the public has access to an assortment of dragonfly-related literature and identification tools [101]. These include published field guides and online portals to submit pictures and sighting metadata [45,78,103,104]. Collectively, the two dragonfly recording initiatives in Africa, the Odonata Database of Africa [105] and OdonataMap [106] host well over 200,000 dragonfly records, with almost 50% being in South Africa. Most records are from Gauteng, Western Cape and KwaZulu-Natal, which are the provinces with the highest level of urbanization in the country (Gauteng and the Western Cape), or has the highest current urbanization rate (KwaZulu-Natal). This means that there is great potential for the use of data acquired through dragonfly citizen science to investigate responses to on-going urbanization [64], or to identify urban areas for restoration or rehabilitation.

## 6. Future Prospects for Dragonfly Research

Changes to spatial and temporal gradients in temperature and water availability as a result of urbanization may lead to overall shifts in dragonfly behavior, including migration, mating, and thermoregulatory activities [107]. While habitat destruction and/or alteration may accelerate local extinction rates for some habitat-specific dragonflies, some species may benefit greatly from artificial freshwater habitats and urban heat island effects [108]. Some dragonflies may be more tolerant of urban impacts and possess physiological, phenological and morphological traits, which allow them to occupy urban environments [109,110,111]. More information on which dragonfly species benefit from freshwater habitats in urban settings, and which species suffer as a result of urbanization is required, so their conservation needs can be determined to prevent local and regional extinctions [64].

Dragonflies are apex predators in the freshwater insect realm, and offer important ecological services such as biological pest control, particularly where disease vectors such as mosquitoes are abundant [101,112]. Dragonflies may also cause disservices by feeding on important pollinators in urban and suburban settings [113,114,115]. As a result, they have a significant influence on freshwater trophic interactions and energy flows between terrestrial and aquatic environments [116]. However, little empirical information is available on shifts in trophic flows in response to urbanization [59].

## 7. Conclusions

The inevitability of increasing urbanization has far reaching consequences for dragonflies through direct and indirect impacts, especially in combination with global climate change. Some aspects of dragonflies in African urban environments are still under-investigated, but by drawing from other continents, which have been confronted with urbanization much earlier, we can act rapidly and effectively to protect these iconic freshwater insects from future population extinctions across Africa. Certain dragonflies are well-equipped to deal with some of the physical and chemical changes brought about by urbanization. Urban landscapes may also create unique opportunities for dragonflies, and urban waterbodies with other primary functions may secondarily increase area of occupancy for some dragonflies. Yet, freshwater habitats in urban areas are only effective reserves for a variety of dragonflies and other aquatic insects when they resemble natural freshwater habitats as closely as possible. This is often only achievable by practicing appropriate management around freshwater habitats (e.g., removing woody invasive alien plants, maintaining water flow, and mitigating pollution) in close association with urban areas. Community outreach and raising public awareness can further assist in securing high-quality habitats for dragonflies, especially in developing countries where the need for freshwater resources are increasing dramatically.

## Figures and Tables

**Figure 1 insects-12-00190-f001:**
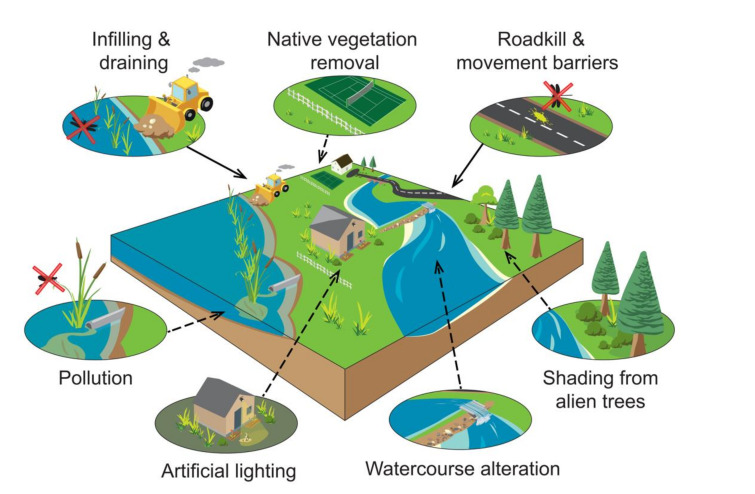
Direct (solid arrows) and indirect (dashed arrows) impacts of urbanization on dragonflies. These impact act synergistically, and together exert immense pressure on dragonflies in urban areas. Red crosses indicate elimination of dragonflies.

**Figure 2 insects-12-00190-f002:**
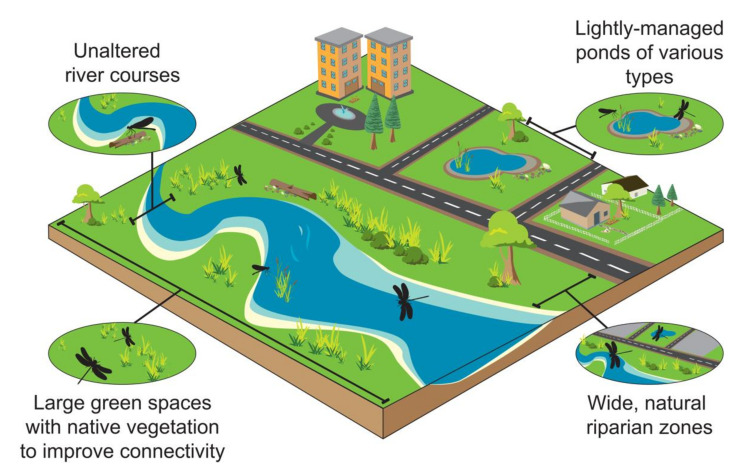
Green and blue spaces provide several opportunities for dragonflies to establish and persist in urban areas.

**Figure 3 insects-12-00190-f003:**
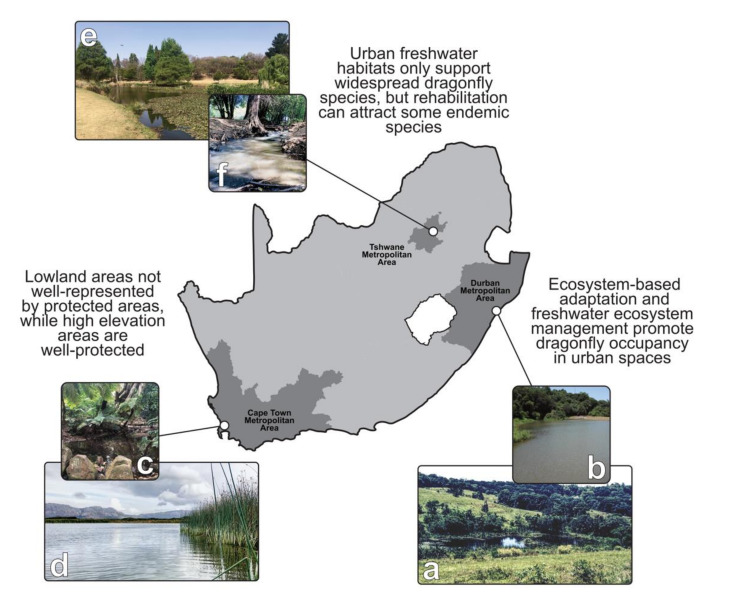
The three largest metropolitan areas in South Africa present different challenges and opportunities for dragonfly conservation. (**a**) A former sugarcane field now part of the Kenneth Stainbank Nature Reserve, (**b**) a dragonfly conservation pond in Pietermaritzburg Botanical Gardens, (**c**) a high elevation mountain stream within the Table Mountain National Park, (**d**) a heavily impacted wetland in the lowlands of the Cape Floristic Region, (**e**) an urban lake in the heart of the Tshwane Metropolitan Area, formerly used as an irrigation reservoir, and (**f**) an impacted stream in a suburban area now being rehabilitated.

**Figure 4 insects-12-00190-f004:**
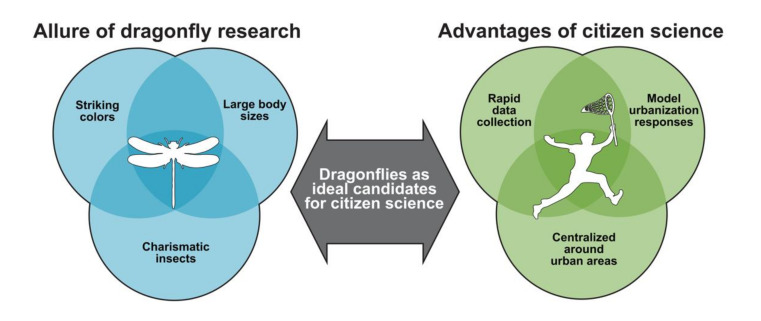
Dragonflies are useful as bioindicators and have high public appeal. Public involvement is key to addressing the conservation needs of dragonflies and other co-occurring insects in urban areas.

## Data Availability

No new data were created or analyzed in this study. Data sharing is not applicable to this article.

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
