# Peer review of "A Review of the Impacts and Opportunities for African Urban Dragonflies"

_insects, 2021, doi:10.3390/insects12030190_

Round 1
Reviewer 1 Report
The manuscript entitled “A review of the impacts and opportunities for African urban dragonflies” by Deacon & Samways is a valuable work reviewing the impact of various factors related to urbanization on dragonflies and precautions for their conservation in urban areas, with a particular focus on southern African cities. The manuscript is well written and linear. In my view, all the topics related to the review subject were addressed, so I have only a few minor suggestions for authors.
I suggest authors to check species/genera names throughout the manuscript since some of them are not italicized, e.g., line 149, 178, 185.
Paragraphs numbers should be modified, there are two different paragraphs numbered as “4” so also the following paragraphs numbering is wrong.
Lines 304-305: The meaning of this sentence is not clear to me, please contextualize it. It sounds more like a title.
Line 310-312: if the total number of dragonflies species present in the area is known, I suggest to mention it here. If I’m not mistaken, in the manuscript you never indicate the total number of species reported for this and for the other area you mention. It would also be interesting for making a comparison with the number of endemic species.
If some species numbers are available also for other urban areas of South Africa I suggest to report them.
Author Response
I suggest authors to check species/genera names throughout the manuscript since some of them are not italicized, e.g., line 149, 178, 185.
Response: We thank the reviewer for pointing this out. Formatting must have changed when we transposed our manuscript to the journal template. We carefully went through the entire manuscript to ensure that all species names are italicized.
Paragraphs numbers should be modified, there are two different paragraphs numbered as “4” so also the following paragraphs numbering is wrong.
Response: We thank the reviewer for pointing out the issue around numbering of sections. The section numbers have now been updated.
Lines 304-305: The meaning of this sentence is not clear to me, please contextualize it. It sounds more like a title.
Response: This sentence has been revised, now reading: ‘Biodiversity patterns are complex in Cape Town, a coastal city built around Table Mountain in the heart of the CFR.’ (L354)
Line 310-312: if the total number of dragonflies species present in the area is known, I suggest to mention it here. If I’m not mistaken, in the manuscript you never indicate the total number of species reported for this and for the other area you mention. It would also be interesting for making a comparison with the number of endemic species.
If some species numbers are available also for other urban areas of South Africa I suggest to report them.
Response: We agree with the reviewer on this comment, some numbers on dragonfly species count would be useful. We have now included numbers on total species richness and endemic species richness for the three case study areas of South Africa (L337-338, 361-362, and 392-393).
Reviewer 2 Report
Summary: This manuscript discusses well-understood effects of urbanization on freshwater resources – with a specific emphasis on dragonflies. Mitigation and conservation approaches are discussed and a few examples of how these measures were applies to specific locales in S. Africa are discussed. Finally, the potential for citizen science to engage in dragonfly conservation is discussed. In reading the manuscript, I was struck by how little I learned. I found that despite the manuscript being very well referenced, and a topic that I am interested in, there was very little specific content presented and I found it to be overly generalized. It is not a review in the sense of pulling together the current state of detailed knowledge to provide a path forward and perspectives for research scientists. Perhaps the lessons learned from conservation efforts could be repackaged for a conservation journal, but as a research scientists, I found very little here that could inform my future efforts.
The manuscript is also marred by several awkward and nonsensical sentences:
Line 23-24: is this grammatically correct?
Line 60: missing “the”
Line 61-63: I’m not sure that a lack of knowledge exacerbates impacts….it is the actions that stem from lack of knowledge that do.
Line 65: what is a “setting to biodiversity” ?
Line 72-73: Ecological impacts are due to economic attractiveness? This makes no sense.
Line 74-77: very awkward sentence construction
Line 90-91: how could this possibly be true?
Line 109-111; what does this sentence mean?
Line 147-148: some specific details here would be useful.
Line 155 and elsewhere: the use of the word “instigation” doesn’t fit
Line 188: you can’t say that things act synergistically without providing specific evidence.
Line 189 – how does this sentence relate to the topic sentence?
Line 237: Is instigated the correct term?
Line 362-363: Relationships are – correct poor grammar
Author Response
Summary: This manuscript discusses well-understood effects of urbanization on freshwater resources – with a specific emphasis on dragonflies. Mitigation and conservation approaches are discussed and a few examples of how these measures were applies to specific locales in S. Africa are discussed. Finally, the potential for citizen science to engage in dragonfly conservation is discussed. In reading the manuscript, I was struck by how little I learned. I found that despite the manuscript being very well referenced, and a topic that I am interested in, there was very little specific content presented and I found it to be overly generalized. It is not a review in the sense of pulling together the current state of detailed knowledge to provide a path forward and perspectives for research scientists. Perhaps the lessons learned from conservation efforts could be repackaged for a conservation journal, but as a research scientists, I found very little here that could inform my future efforts.
Response: We are very sorry to hear that our manuscript is of little interest to anonymous reviewer 2. We can only speculate that this particular reviewer is a well-established scientist in the field, hence our extra careful consideration of all comments provided here. We would also like to take the opportunity to thank the reviewer for the valuable comments provided.
We would like to point out that almost all organisms, whether insects, birds, or mammals, face similar threats against the background of urban expansion. They are also presented with similar opportunities, and mitigation measures in place to conserve one group will almost certainly aid the conservation of another. Most urban threats which endanger biodiversity have manifested over many years, and most urban ecologists know what these threats are. This challenges our ability to present insights on new threats, especially so for dragonflies in South Africa.
We had three main objectives with this review:
1) to collate existing and updated global information on the threats to dragonflies in urban spaces, including those which are perhaps not well known such as artificial lighting, roads, etc. We wanted to present this in a simple format that is easy to understand, even for scientists outside of urban ecology and civil society.
2) to summarize existing global information on the opportunities that urban spaces can provide to dragonflies when they are managed properly, given that urbanization is inevitable and has manifested over a very long time. This objective was grounded on the fact that human wellbeing and freshwater conservation in urban areas go hand-in-hand. Again, we felt that it was important to keep the format simple enough to be more inclusive of a wider audience.
3) Finally, we placed three major South African urban areas in the global context to identify specific threats, and to highlight which conservation measure have succeeded and which have not succeeded to protect the rich dragonfly fauna of the country.
To the best of our knowledge, this is a first attempt to do so for South Africa. Urban ecology is still in it’s infancy in South Africa (and in Africa), challenging our ability to present much information on new threats. We hope that this manuscript will spike an interest in urban research in South Africa (a field which is greatly under-represented in the country) and other countries which face a dearth of information on urban impacts. We hope that we can gain new insights in future into the impacts of urban areas in this truly unique country with its complex climatic gradients, highly variable topography, and interesting socio-economic systems.
The manuscript is also marred by several awkward and nonsensical sentences:
Line 23-24: is this grammatically correct?
Response: This sentence was revised to read: ‘Urbanization presents many challenges to maintenance of freshwater quality and conservation of freshwater biota, especially in Africa.’ (L23-24).
Line 60: missing “the”
Response: ‘the’ was added before ‘space limitations’ (L64).
Line 61-63: I’m not sure that a lack of knowledge exacerbates impacts….it is the actions that stem from lack of knowledge that do.
Response: This sentence was revised, now reading: ‘Ecological impacts are expected to be particularly severe in ecologically sensitive areas, further exacerbated by ineffective environmental regulations due to limited scientific knowledge on biological responses to urbanization.’ (L66-68).
Line 65: what is a “setting to biodiversity” ?
Response: the word ‘settings’ was replaced by the word ‘conditions’ to improve on clarity (L69).
Line 72-73: Ecological impacts are due to economic attractiveness? This makes no sense.
Response: This sentence was removed, and instead the first sentence of the paragraph was updated to read: ‘Within sub-Saharan Africa, southern Africa is the most urbanized region due to its accessible coastline, abundant mineral deposits and rich soils’ (L72-73).
Line 74-77: very awkward sentence construction
This section was revised, now reading ‘Most aquatic insect groups are poorly studied across the continent. Southern African dragonflies (referring to both dragonflies and damselflies; Odonata: Anisoptera and Zygoptera) are by far the most studied group of insects in terms of their taxonomy and biogeography (L79-82).
Line 90-91: how could this possibly be true?
Response: We understand the confusion regarding this statement. This sentence was revised to read ‘…with losses in freshwater habitats strongly correlated with losses of green space in some countries.’ (L115).
Line 109-111; what does this sentence mean?
Response: to avoid confusion, this section was reworded to read: ‘This is a particular problem for small natural ponds in southern Africa, many of which are regarded as ecologically unimportant. These small freshwater habitats are in fact highly complementary at the regional scale, and some are important habitats for rare dragonflies.’.
Line 147-148: some specific details here would be useful.
Response: This sentence was revised, now reading: ‘In addition to shading, alien vegetation encroachment also contributes to soil erosion through homogenization of riparian plant communities, so leading to increased runoff into rivers and higher sediment loads.’ (L179-181). We’ve also added to L 185-186 ‘(e.g. through reduction in oxygen supplies and increases in water temperature)’.
Line 155 and elsewhere: the use of the word “instigation” doesn’t fit
Response: The first instance of ‘instigation’ was replaced by ‘construction’ (L 188).
Line 188: you can’t say that things act synergistically without providing specific evidence.
Response: This sentence was revised to read ‘Potential urban impacts on dragonflies and other aquatic macroinvertebrates vary with context.’
Line 189 – how does this sentence relate to the topic sentence?
Response: To avoid confusion, we removed the mention of ‘synergy’ or ‘acting together’ from the leading sentence. This paragraph now only states how different pond contexts shape dragonfly assemblages (L230-243).
Line 237: Is instigated the correct term?
The second instance was replaced by ‘built’ (L276).
Line 362-363: Relationships are – correct poor grammar
Response: We thank the reviewer for pointing this out. ‘is’ was replaced with ‘are’ (L418).
Reviewer 3 Report
This is nice review study.
I would like to see more about artificial ponds without fishes. The comparison between ponds with fishes and with out fishes. How fish/no-fish ponds affect dragonfly diversity and conservation in urban environment. This aspect is only missing part in this review.
Minor comments:
I found some scientific names which is not in italic.
Author Response
This is nice review study.
I would like to see more about artificial ponds without fishes. The comparison between ponds with fishes and with out fishes. How fish/no-fish ponds affect dragonfly diversity and conservation in urban environment. This aspect is only missing part in this review.
Response: We thank the reviewer for the kind compliment. Fish introductions in urban (or at least peri-urban) areas is a big concern for aquatic macroinvertebrates in general. A few old studies have indicated that Rainbow trout (Oncorhynchus mykiss) and Largemouth bass (Micropterus salmoides) are the most prolific invaders of dams and rivers, yet very few recent studies (especially in Africa) have drawn comparison between stocked and fishless ponds. With regards to the ornamental fish in urban areas, Koi (Cyprinus rubrofuscus) and Goldfish (Carassius auratus), they pose probably only minor risk to aquatic macroinvertebrates in urban areas.
Nevertheless, to acknowledge that fish may have some impact on dragonflies in urban spaces, we expanded on L 197-200, now reading ‘Damming of rivers also introduces other disturbances, such as alien fish (e.g. Oncorhynchus mykiss and Micropterus salmoides) for recreational fishing. These generalist predators feed in different areas of the water column, so leading to population declines of dragonfly larvae and adults.’.
Minor comments:
I found some scientific names which is not in italic.
Response: We thank the reviewer for pointing this out. Formatting must have changed when we transposed our manuscript to the journal template. We carefully went through the entire manuscript to ensure that all species names are italicized.